# A Closer Look at the Adversarial Robustness of Deep Equilibrium Models

**Zonghan Yang**[1], **Tianyu Pang**[2], **Yang Liu**[1,3,4*]
[1]Department of Computer Science and Technology, Tsinghua University, Beijing, China
[2]Sea AI Lab, Singapore
[3]Institute for AI Industry Research (AIR), Tsinghua University, Beijing, China
[4]Beijing Academy of Artificial Intelligence, Beijing, China
`yangzh20@mails.tsinghua.edu.cn`, `tianyupang@sea.com`, `liuyang2011@tsinghua.edu.cn`

## Abstract

Deep equilibrium models (DEQs) refrain from the traditional layer-stacking paradigm and turn to find the fixed point of a single layer. DEQs have achieved promising performance on different applications with featured memory efficiency. At the same time, the adversarial vulnerability of DEQs raises concerns. Several works propose to certify robustness for monotone DEQs. However, limited efforts are devoted to studying empirical robustness for general DEQs. To this end, we observe that an adversarially trained DEQ requires more forward steps to arrive at the equilibrium state, or even violates its fixed-point structure. Besides, the forward and backward tracks of DEQs are misaligned due to the black-box solvers. These facts cause gradient obfuscation when applying the ready-made attacks to evaluate or adversarially train DEQs. Given this, we develop approaches to estimate the intermediate gradients of DEQs and integrate them into the attacking pipelines. Our approaches facilitate fully white-box evaluations and lead to effective adversarial defense for DEQs. Extensive experiments on CIFAR-10 validate the adversarial robustness of DEQs competitive with deep networks of similar sizes.

## 1 Introduction

Conventional deep networks employ multiple stacked layers to process data in a feedforward manner [17]. During training, network parameters are optimized by backpropagating loss updates through the consecutive layers [36]. Recently, [3] propose deep equilibrium models (DEQs), whose forward pass involves finding the fixed point (i.e., equilibrium state) of a single layer. With implicit differentiation, the backward pass of DEQs is formulated as another linear fixed-point system. Training DEQs with black-box root solvers only consumes $\mathcal{O}(1)$ memory, which enables DEQs to achieve performance competitive with conventional networks in large-scale applications, including language modelling [3], image classification and segmentation [4], density modelling [24, 16], and graph modelling [23].

Considering the fixed point as a local attractor, DEQs are expected to be stable to small input perturbations. However, empirical observations show the opposite that a vanilla DEQ is also vulnerable to adversarial attacks [16]. Along this routine, several works are proposed to investigate the certified robustness for monotone DEQs [40, 34, 27, 20, 28, 10]. Inspired from the monotone operator splitting theories, monotone DEQs are designed with the guarantee of existence and convergence of equilibrium points. However, the layer parameterization of monotone DEQs and the limited scalability of certification methods narrow the scope of these previous studies. On the other hand, [16] explore the adversarial robustness for general DEQs. They incorporate the adversarial generation process into the equilibrium solver to accelerate the PGD attack [25]. Nevertheless, the PGD attack is originally designed for deep networks, requiring for end-to-end white-box differentiation. In contrast, DEQs

---

[*]Corresponding author: Yang Liu

36th Conference on Neural Information Processing Systems (NeurIPS 2022).

rely on black-box solvers and could obfuscate the gradients used in PGD: as shown in Fig. 2-(a), in DEQs trained with different configurations, the intermediate states *always* exhibit higher robustness than the final state under *ready-made* PGD attacks. Compared to the extensive literature on the adversarial robustness of deep networks [6, 38, 15, 26, 22, 25, 42, 35, 29], much less is known about the adversarial robustness of general DEQs, especially under a well-elaborate white-box setting. This motivates us to disentangle the modules in DEQs and provide a fair evaluation of their robustness.

In this paper, we first summarize the challenges of training robust DEQs (see Sec. 3), including (**i**) convergence of the black-box solvers and (**ii**) misalignment between the forward and backward passes. The off-the-shelf attacks work in a gray-box setting as they have no access to the intermediate states in the forward pass. To thoroughly evaluate the robustness, we propose two methods for intermediate gradient estimation: the first one is iterating adjoint gradient estimations simultaneously in the forward pass, as formally described in Sec. 4.1; the second one is estimating intermediate gradients by unrolling, as seen in Sec. 4.2. Then in Sec. 5, we develop approaches to integrate the estimated gradients into the ready-made attacks towards fully white-box adversaries. We also design defense strategies for DEQs to boost their robustness under white-box attacks.

We use PGD-AT to train large-sized and XL-sized DEQs on CIFAR-10. To benchmark their robustness [12], the parameter sizes of the DEQs are set to be comparable with ResNet-18 [18] and WideResNet-34-10 [41], respectively. We observe that the adversarially trained DEQs with the exact gradient [3] require more forward steps to arrive at the equilibrium state, or even violate their fixed-point structures. We also find an intriguing robustness accumulation effect that the intermediate states in the forward pass are more robust under ready-made attacks. These phenomena exhibit gradient obfuscation [2], which verifies the necessity of intermediate gradient estimation to construct white-box attacks and defense strategies. Robustness performance under the white-box evaluation shows that DEQs achieve competitive or stronger adversarial robustness than deep networks of similar parameter amounts. Our investigation sheds light on the pros and cons with respect to the adversarial robustness of DEQs.

## 2 Background

This section includes the background on DEQs and adversarial robustness for deep networks.

### 2.1 Deep equilibrium models

We first briefly introduce the modelling of deep equilibrium models (DEQs) [3, 4]. Consider a $T$-layer weight-tied input-injected neural network:

$$\mathbf{z}_n = f_\theta(\mathbf{z}_{n-1}; \mathbf{x}), \ n = 1, \dots, T, \tag{1}$$

where $\mathbf{x} \in \mathbb{R}^l$ is the input, $\mathbf{z}_n \in \mathbb{R}^d$ is the output of the $n$-th layer, and $\theta$ is the network weights shared across different layers. One can cast the evolution of $\{\mathbf{z}_n\}$ as a fixed-point iteration process. When $n \to \infty$, $\mathbf{z}_n$ converges to the fixed point $\mathbf{z}^*$ which satisfies the equation $\mathbf{z}^* = f_\theta(\mathbf{z}^*; \mathbf{x})$.

Deep equilibrium models rely on the fixed-point equation and leverage a black-box solver to *directly* solve for $\mathbf{z}^*$ in the forward pass. The backward pass of DEQs can also be formulated as a fixed-point iteration process. With the loss function $L(\mathbf{z}^*, y)$ and implicit differentiation, we can compute the gradient with respect to $\theta$ or $\mathbf{x}$ with

$$\frac{\partial L}{\partial(\cdot)} = \left(\frac{\partial f_\theta(\mathbf{z}^*; \mathbf{x})}{\partial(\cdot)}\right) \underbrace{\left(I - \frac{\partial f_\theta(\mathbf{z}^*; \mathbf{x})}{\partial \mathbf{z}}\right)^{-1} \frac{\partial L(\mathbf{z}^*, y)}{\partial \mathbf{z}}}_{\mathbf{u}^*}, \tag{2}$$

where $(\partial \mathbf{a}/\partial \mathbf{b})_{ij} = \partial \mathbf{a}_j / \partial \mathbf{b}_i$ and $\mathbf{u}^*$ satisfies

$$\mathbf{u}^* = \left(\frac{\partial f_\theta(\mathbf{z}^*; \mathbf{x})}{\partial \mathbf{z}}\right) \mathbf{u}^* + \frac{\partial L(\mathbf{z}^*, y)}{\partial \mathbf{z}}. \tag{3}$$

According to Eq. (3), the backward pass can also be executed with a black-box fixed-point solver, and this iteration process is independent of that in the forward pass.

Several techniques have been proposed to improve the training stability of DEQs. [5] propose to regularize the Jacobian matrix in Eq. (2) during training so that the nonlinear forward system and the backward linear system enjoy appropriate contractivity. [14] propose unrolling-based and

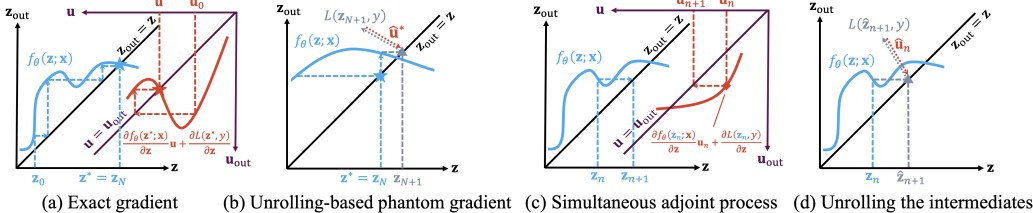

(a) Exact gradient    (b) Unrolling-based phantom gradient    (c) Simultaneous adjoint process    (d) Unrolling the intermediates

Figure 1: The gradients proposed for DEQs. (a): the exact gradient [3] solved by an independent fixed-point iteration process. (b): the unrolling-based phantom gradient [14] returned by automatic differentiation on a computational subgraph where the equilibrium state $\mathbf{z}^*$ is unrolled. (c): simultaneous adjoint process along with the forward iterations described in Sec. 4.1. (d): unrolling the intermediate states $\mathbf{z}_n$ for gradient estimation in Sec. 4.2. We leverage (c) and (d) to estimate intermediate gradients and design fully white-box attacks to evaluate the robustness of DEQs.

Neumann-series-based phantom gradients to replace the exact gradient in Eq. (2) for acceleration. The unrolling-based phantom gradient is defined as

$$
\lambda \sum_{t=1}^{k} \left( \frac{\partial f_\theta \left( \hat{\mathbf{z}}_{N+t}; \mathbf{x} \right)}{\partial(\cdot)} \right) \mathbf{P}_{\lambda, \mathbf{z}_N}^{(t)} \frac{\partial L(\hat{\mathbf{z}}_{N+k}, y)}{\partial \mathbf{z}}, \tag{4}
$$

where

$$
\mathbf{P}_{\lambda, \mathbf{z}_N}^{(t)} = \prod_{s=t+1}^{k} \left( \lambda \frac{\partial f_\theta \left( \hat{\mathbf{z}}_{N+s}; \mathbf{x} \right)}{\partial \mathbf{z}} + (1-\lambda)I \right), \tag{5}
$$

$$
\hat{\mathbf{z}}_{N+t} = (1-\lambda)\hat{\mathbf{z}}_{N+t-1} + \lambda f_\theta \left( \hat{\mathbf{z}}_{N+t-1}; \mathbf{x} \right) \tag{6}
$$

are the $k$ unrolling steps with $1 \leq t \leq k$, starting from $\hat{\mathbf{z}}_N = \mathbf{z}^*$ returned by the forward solver.

Eq. (4) is calculated by the automatic differentiation framework [30] on the computational subgraph in Eq. (6). It is demonstrated that the unrolling-based phantom gradient imposes implicit Jacobian regularization effect to DEQ training [14]. DEQs trained by either the exact or the phantom gradients are competitive to deep neural networks in terms of natural accuracy. In our work, we leverage adversarial defense strategies to train DEQs to improve their robustness.

## 2.2 Adversarial robustness for deep networks

Much research has been dedicated to adversarial attacks and defenses of deep neural networks. On the one hand, white-box adversarial attack techniques like PGD [25] construct adversaries by iteratively perturbing inputs in the gradient ascent direction. The robustness of deep networks is benchmarked by AutoAttack [12], which consists of four attacks including two PGD variants with adaptive stepsize and the query-based SQUARE attack [1]. On the other hand, adversarial training (AT) [25] is one of the most effective defense strategies. By early stopping the training procedure as in [35], the primary PGD-AT framework still achieves competitive robustness performance compared with the state-of-the-art defense techniques like TRADES [42]. It is worth mentioning that many defense approaches claim robustness improvement by obfuscating gradients, which proves to be a false sense of security under adaptive attacks designed specifically [2]. In our work, we train DEQs with PGD-AT and investigate their adversarial robustness by designing customized defenses and adaptive attacks.

## 3  Challenges for robust general DEQs

This section describes the challenges encountered when we aim to train robust general DEQs.

**Misalignment between forward & backward passes.** The central idea of DEQs is *directly* solving for the equilibrium state $\mathbf{z}^*$ and differentiating through the fixed point equation $\mathbf{z}^* = f_\theta(\mathbf{z}^*; \mathbf{x})$ for efficient forward and backward passes. Fig. 1-(a) sketches the calculation of the exact gradient [3]. Independent from the forward iterations (the **blue** curve), the exact gradient is acquired by solving for a linear fixed-point system that only depends on the equilibrium state $\mathbf{z}^*$ (the **orange** curve). Fig. 1-(b) shows the calculation of the unrolling-based phantom gradient [14]. $\mathbf{z}^*$ as the final state in the forward pass is unrolled (the **gray** iteration), and the gradient is obtained from the automatic differentiation on the loss function. However, when iterating the gradient computations,

the intermediate states $\{\mathbf{z}_n\}$ in the forward pass are bypassed by both methods. The misalignment between the forward and backward tracks results in a gray-box setting for the ready-made attacks.

**Convergence of the black-box solvers.** In contrast with monotone DEQs, there is no guarantee for the existence and convergence of the equilibrium states in general DEQs. It is thus unknown whether the black-box solvers in DEQs still converge to equilibrium states under input perturbations. Adversarial training also adds the concern on equilibrium convergence. The well-known effect of adversarial training for deep networks is the trade-off between robustness and accuracy [37, 39, 42, 32]. A similar drop in standard accuracy (from $78\%$ to $55\%$) is also observed for tiny-sized adversarially-trained DEQs [16]. The robustness-accuracy trade-off brings training instability for general DEQs, which may take more iterations in the solvers for equilibrium convergence, or even violate their fixed-point structures. Finally, the robustness comparison is still under-explored between large-sized general DEQs and deep networks with similar parameter counts.

## 4  On intermediate gradient estimation

As the forward and backward tracks in DEQs are misaligned, the intermediate states in the forward pass are inaccessible to off-the-shelf attacks, which causes gradient obfuscation and results in false positive robustness. Therefore, it is necessary to estimate the intermediate gradients. With the integration of the estimated gradients, the attacks can validate the robustness of DEQs in a fully white-box setting. In this section, we propose two methods for intermediate gradient estimation.

### 4.1  Simultaneous adjoint in the forward pass

Inspired by the adjoint process in neural ODE models [9], we propose the adjoint process for intermediate gradient estimation in DEQs. The adjoint process in neural ODE models is characterized by an adjoint ODE [31]. For DEQs, we propose to iterate the updates of adjoint states subject to $\mathbf{z}_n$ in the forward pass. We investigate the simultaneous adjoint with Broyden's method [7] as the forward solver. In the forward pass, Broyden's method updates the intermediate state $\mathbf{z}_n$ based on the residual $g_\theta(\mathbf{z}_n; \mathbf{x}) = f_\theta(\mathbf{z}_n; \mathbf{x}) - \mathbf{z}_n$ and $B_n$, the low-rank approximation of the Jacobian inverse:

$$\mathbf{z}_{n+1} = \mathbf{z}_n - \alpha B_n g_\theta(\mathbf{z}_n; \mathbf{x}), \ \mathbf{z}_0 = \mathbf{0} \tag{7}$$

$$B_{n+1} = B_n + \frac{\Delta \mathbf{z}_{n+1} - B_n \Delta g_{n+1}}{\Delta \mathbf{z}_{n+1}^{\mathrm{T}} B_n \Delta g_{n+1}} \Delta \mathbf{z}_{n+1}^{\mathrm{T}} B_n, \tag{8}$$

where $0 \leq n \leq N-1$, $B_0 = -I$, $\Delta \mathbf{z}_{n+1} = \mathbf{z}_{n+1} - \mathbf{z}_n$, $\Delta g_{n+1} = g_\theta(\mathbf{z}_{n+1}; \mathbf{x}) - g_\theta(\mathbf{z}_n; \mathbf{x})$, and $\alpha$ is the step size. To maintain a simultaneous adjoint, we start from $\mathbf{u}_0 = \mathbf{0}$ and use Broyden's method to solve Eq. (3). Similar with the residual function $g_\theta(\cdot; \mathbf{x})$ for $\mathbf{z}_n$, the fixed-point equation in Eq. (3) defines the residual of the adjoint state. However, we propose to replace the $\mathbf{z}^*$ in Eq. (3) by $\mathbf{z}_n$, and integrate the approximated Jacobian inverse $B_n$ to force the alignment of the adjoint state update:

$$\mathbf{v}_n = \left( \frac{\partial f_\theta(\mathbf{z}_n; \mathbf{x})}{\partial \mathbf{z}} \right) \mathbf{u}_n + \frac{\partial L(\mathbf{z}_n, y)}{\partial \mathbf{z}} - \mathbf{u}_n, \tag{9}$$

$$\mathbf{u}_{n+1} = \mathbf{u}_n - \beta B_n \mathbf{v}_n, \tag{10}$$

where $\mathbf{v}_n$ is the residual at iteration $n$, $\mathbf{u}_n$ is the updated adjoint state, and $\beta > 0$ is the step size.

We use the following surrogate gradients to construct attacks on the intermediate state $\mathbf{z}_n$:

$$\left[ \widetilde{\frac{\partial L}{\partial x}} \right]_n = \left( \frac{\partial f_\theta(\mathbf{z}_n; \mathbf{x})}{\partial \mathbf{x}} \right) \mathbf{u}_n. \tag{11}$$

An illustration for the simultaneous adjoint process is shown in Fig. 1-(c). In the following, we refer to this method as *simultaneous adjoint* when constructing intermediate state attacks in Sec 5.1.

*Remark* 4.1. We show in Appendix B that under mild assumptions, the $\{\mathbf{u}_n\}$ converges to $\mathbf{u}^*$ when $0 < \beta < 1$ in Eq. (10). However in practice, we do *not* require the convergence of $\mathbf{u}_n$ as we only use them in Eq. (11) as gradient *estimations* to construct *intermediate attacks*.

*Remark* 4.2. Similar with the update of $\mathbf{u}_n$ in our approach, [16] propose augmented DEQs as an integration of the iterative updating process of $\mathbf{z}$, $\mathbf{u}$, and $\mathbf{x}$ as a whole:

$$F \left( \begin{bmatrix} \mathbf{z}_n \\ \mathbf{u}_n \\ \mathbf{x}_n \end{bmatrix} \right) = \begin{bmatrix} f_\theta(\mathbf{z}_n; \mathbf{x}_n) \\ \left( \frac{\partial f_\theta(\mathbf{z}_n; \mathbf{x}_n)}{\partial \mathbf{z}} \right) \mathbf{u}_n + \frac{\partial L(\mathbf{z}_n, y)}{\partial \mathbf{z}} \\ \mathbf{x}_n - \left( \frac{\partial f_\theta(\mathbf{z}_n; \mathbf{x}_n)}{\partial \mathbf{x}} \right) \mathbf{u}_n \end{bmatrix} \tag{12}$$

The augmented DEQs leverage a black-box solver (e.g., Broyden's method) to find the equilibrium of the whole state $[\mathbf{z}^*, \mathbf{u}^*, \mathbf{x}^*]^{\mathrm{T}}$. However, several cross-terms exist in the joint Jacobian due to the coupling of the three iteration processes, which further hinders the convergence. In contrast, the simultaneous adjoint update in Eq. (10) does not include the update of $\mathbf{x}$. Furthermore, we reuse the Jacobian inverse approximation matrix $B_n$ in the update of $\mathbf{u}_n$, which is easy to implement. Because of the disentanglement of $\mathbf{z}_n$ and $\mathbf{u}_n$ in the joint Jacobian, our method also enjoys better efficiency and flexibility as one can early exit the adjoint process without affecting the updates of $\mathbf{z}_n$'s.

*Remark* 4.3. Concurrent work [33] also explores the idea of sharing approximated Jacobian inverse $B_n$ in bi-level optimization problems. While their motivation is to accelerate DEQ training, we use the adjoint states to construct gradient estimation and facilitate white-box attacks. We also compare our work with theirs in terms of intermediate state attacks; See Appendix D for details.

### 4.2 Unrolling the intermediate states

We also propose to estimate the gradient at the state $\mathbf{z}_n$ by unrolling. Depicted in Fig. 1-(d), $\mathbf{z}_n$ is involved in an artificially constructed computational graph. We can thus estimate the intermediate gradient by backpropagation with automatic differentiation. Formally, applying Eq. (4) to $\mathbf{z}_n$ yields

$$\left[ \widehat{\frac{\partial L}{\partial \mathbf{x}}} \right]_n^{(k)} = \mathbf{A}_{\lambda, \mathbf{z}_n}^{(k)} \frac{\partial L(\hat{\mathbf{z}}_{n+k}, y)}{\partial \mathbf{z}}, \tag{13}$$

where

$$\mathbf{A}_{\lambda, \mathbf{z}_n}^{(k)} = \lambda \sum_{t=0}^{k-1} \left( \frac{\partial f_\theta(\hat{\mathbf{z}}_{n+t}; \mathbf{x})}{\partial \mathbf{x}} \right) \mathbf{P}_{\lambda, \mathbf{z}_n}^{(k)}, \tag{14}$$

$$\mathbf{P}_{\lambda, \mathbf{z}_n}^{(k)} = \prod_{s=t+1}^{k-1} \left( \lambda \frac{\partial f_\theta(\hat{\mathbf{z}}_{n+s}; \mathbf{x})}{\partial \mathbf{z}} + (1 - \lambda)I \right), \tag{15}$$

and the state sequence $\hat{\mathbf{z}}_n, \hat{\mathbf{z}}_{n+1}, \cdots, \hat{\mathbf{z}}_{n+k}$ represents the damped unrolling iteration:

$$\hat{\mathbf{z}}_{n+t} = (1 - \lambda)\hat{\mathbf{z}}_{n+t-1} + \lambda f_\theta(\hat{\mathbf{z}}_{n+t-1}; \mathbf{x}), \tag{16}$$

with $t = 1, 2, \cdots, k$ and $\hat{\mathbf{z}}_n = \mathbf{z}_n$. While Eq. (4) is proposed as an approximation of the exact gradient, we unroll the states $\mathbf{z}_n$ for intermediate gradient estimation. Similar to the case of Eq. (11), we use Eq. (13) as estimation to design intermediate attacks for DEQs. We refer to this method as *unrolled intermediates* in the following when incorporating Eq. (13) into the white-box attacks.

## 5 White-box attacks and defenses for DEQs

This section describes different types of white-box attacks and defense strategies for DEQs.

### 5.1 White-box attacks for DEQs

The existing attacks leverage the gradients calculated at the *final* state outputted by the forward solver. Based on the surrogate intermediate gradients in Eq. (11) or Eq. (13), we can involve the $\mathbf{z}_n$ in the forward pass into the construction of adversaries. A direct white-box approach is to use the estimated gradient at an *early* state $\mathbf{z}_n$ as an alternative for input perturbations. Another simple yet effective method is to average the intermediate gradients as the gradient ensemble for attacks. For example, the average of all intermediate gradients along the simultaneous adjoint process is given by

$$\sum_n \left[ \widehat{\frac{\partial L}{\partial \mathbf{x}}} \right]_n = \sum_n \left( \frac{\partial f_\theta(\mathbf{z}_n; \mathbf{x})}{\partial \mathbf{x}} \right) \mathbf{u}_n. \tag{17}$$

The gradient ensemble can be viewed as the fusion of all perturbation directions indicated by all $\mathbf{z}_n$'s.

### 5.2 Defenses with intermediate states

In addition to the final state $\mathbf{z}^*$, the unused intermediate states can be leveraged as well for the defenses of DEQs. A simple yet effective defense strategy is to *early exit* the forward solver during

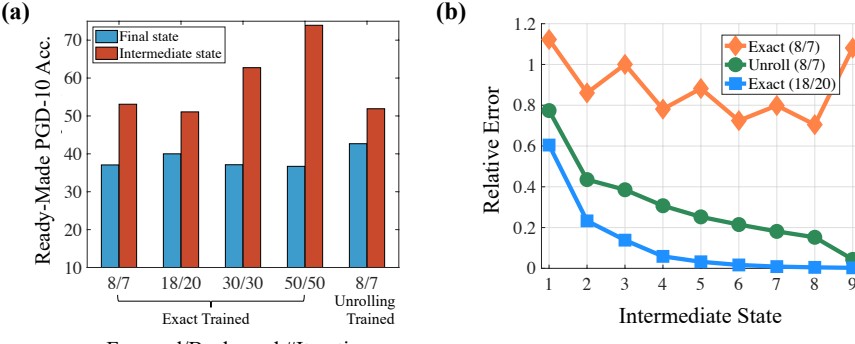

Figure 2: Challenges in benchmarking adversarial robustness of DEQs. (a) Gradient obfuscation issues arise in the DEQs trained with different configurations. With different iteration settings in the DEQ solver or different gradient formulations, the intermediate state *always* exhibit higher robustness than the final state under ready-made PGD-10 attack. (b) Exact-trained DEQ with small iterations violate the fixed-point structure and require more iterations to retain it (analyzed in Sec. 6.1). Both observations motivate us to design adaptive attacks for white-box robustness evaluation for DEQs.

inference. We can evaluate the robustness of DEQs with the early state $\mathbf{z}_n$ in the forward pass, as $\mathbf{z}_n$ and $\mathbf{z}^*$ have the same shape. We determine the optimal timing for early exit by selecting the top robustness performance of all $\mathbf{z}_n$'s on the development set under the ready-made PGD-10 attack.

The input-injected neural network provides an interpretation for the early-state defense. From Eq. (1), the distortion of $\mathbf{z}_n$ comes from both the perturbed $\mathbf{z}_{n-1}$ and the biased transformation $f_\theta(\cdot; \mathbf{x}+\Delta\mathbf{x})$. By early exiting the forward process, one obtains a less distorted intermediate state.

Another defense strategy for DEQs is leveraging the *ensemble* of intermediate states. Similar with Eq. (17), we average the intermediate states $\{\mathbf{z}_n\}$ to defend against attacks. Instead of early stopping, the intermediate state ensemble exploits the state representations at all iterations in the forward solver.

While the proposed defense techniques leverage the intermediate states, they still require only $\mathcal{O}(1)$ memory. For early state defense, we determine the optimal time to early exit the solver on the development set offline for once and then fix the early exit step during testing. For ensemble state defense, we maintain an accumulator to sum up $\{\mathbf{z}_n\}$ along the forward pass without storing them.

# 6  Experiments

Following the settings in [5], we experiment with the large-sized DEQ with its parameter count similar to ResNet-18 [17]. We also experiment with an XL-sized DEQ with its parameter count similar to WideResNet-34-10 [41] to enable a fair comparison with the empirical robustness of the deep networks. The detailed experimental settings are listed in Appendix A. We first train DEQs on CIFAR-10 [21] with the PGD-AT framework [25], then test the adaptive attacks and defense strategies proposed in Sec. 5 on the adversarially-trained DEQs. We refer to a DEQ as "exact-trained" when using the exact gradient, and "unrolling-trained" when using the unrolling-based phantom gradient in the PGD-AT framework to generate adversaries and optimize for model parameters. Unless specified, all DEQs are adversarially trained in this paper. During training, we use 10-step PGD with the step size of $2/255$ to generate adversaries within the range of $\ell_\infty = 8/255$. For the specific type of attacks, we use PGD and AutoAttack (AA) [13] to instantiate the white-box attacks in Sec. 5.1.

## 6.1  The retention of the fixed-point structure

We start with the observation on the fixed-point structure. Shown in Fig. 2-(b), the lines illustrate the relative error $\|f_\theta(\mathbf{z}_n; \mathbf{x}) - \mathbf{z}_n\|_2 / \|f_\theta(\mathbf{z}_n; \mathbf{x})\|_2$ for each $\mathbf{z}_n$ [2]. We find that for the exact-trained DEQ with small iteration settings (8 forward / 7 backward iterations), all the relative errors are larger than 0.75, i.e., the forward solver in the DEQ fails to converge to an equilibrium. Such phenomenon

---

[2]The 9-th intermediate state comes from the implementation in DEQs. In the exact-trained DEQ, $\mathbf{z}_9 = f_\theta(\mathbf{z}_8; \mathbf{x})$. In the unrolling-trained DEQ, $\mathbf{z}_9$ is the final state after unrolling $\mathbf{z}_8$. For the exact-trained DEQ with 18 forward / 20 backward iterations, we only plot the first 9 states for an easy comparison.

Table 2: Performance (%) of the unrolling-trained DEQ-Large with the small (8/7) iteration setting and the exact-trained DEQ-Large with the large (50/50) iteration setting under PGD-10. The "final" rows and columns represent the original DEQ output and the ready-made attacks at the final state. The "early" rows indicate early state defense, and the "intermediate" columns indicate the performance of the strongest intermediate attacks. The rows and the columns of "ensemble" demonstrate the ensemble defense and the white-box attacks based on gradient ensemble. Under the (underlined) strongest attacks, the ensemble defense achieves the best robustness performance (**in bold**).

| Training Configurations | Defense | Clean | Simultaneous Adjoint | | | Unrolled Intermediates | | |
|---|---|---|---|---|---|---|---|---|
| | | | Final | Intermediate | Ensemble | Final | Intermediate | Ensemble |
| (8/7) Unrolling-Trained | Final | 78.03 | 49.81 | 59.49 | 54.91 | 42.67 | 62.24 | 51.52 |
| | Early | 79.57 | 54.90 | 39.19 | 42.76 | 51.90 | 29.38 | 34.20 |
| | Ensemble | 79.67 | 51.52 | 52.43 | 49.47 | 49.02 | 55.10 | **47.12** |
| (50/50) Exact-Trained | Final | 73.51 | 37.77 | 70.52 | 43.70 | 36.70 | 69.29 | 48.08 |
| | Early | 86.98 | 75.25 | 12.44 | 40.12 | 73.93 | 18.24 | 26.22 |
| | Ensemble | 75.12 | 40.20 | 72.41 | 45.06 | **39.18** | 68.83 | 49.10 |

reflect the challenge on the convergence of the black-box solvers in DEQs mentioned in Section 3. It is shown that a larger iteration setting is required (18 forward / 20 backward iterations) for exact-trained DEQs to retain the fixed-point structure. In contrast, we find the small iteration setting (8/7) is enough for the unrolling-trained DEQ to retain the fixed-point structure.

It is necessary to retain the fixed-point structure, otherwise leading to gradient obfuscation issues. As is derived from the implicit differentiation on the fixed-point equation $\mathbf{z}^* = f_\theta(\mathbf{z}^*; \mathbf{x})$, the exact gradient in Eq. (2) becomes inexact when the equilibrium point $\mathbf{z}^*$ is not reached. Table 1 shows the empirical performance of the exact-trained DEQ under the small (8/7) iteration setting. The severe performance degradation under alternative gradient formulations as well as the SQUARE attack also indicates gradient obfuscation, as suggested in [2] and [8].

While large iterations for the exact-trained DEQs keep the fixed-point structure, it in-

Table 1: Performance (%) of the exact-trained DEQ-Large with the small (8/7) solver iterations under different attacks. The high accuracy under PGD-10 with the exact gradient is deteriorated using the unrolling-based phantom gradient. Leveraging the query-based SQUARE leads to even lower accuracy. These observations indicate that the DEQ with violated fixed-point structure suffers severe robustness degradation.

| Gradient | Clean | PGD-10 | PGD-1000 | SQUARE |
|---|---|---|---|---|
| Exact | 78.24 | 79.97 | 80.10 | 5.95 |
| Unrolling | | 37.07 | 36.39 | |

evitably slows down the training speed (detailed in Appendix E.3). For the exact-trained DEQs with the small (8/7) iteration setting, we have also tried with varied Jacobian regularization weights to impose stricter Lipschitz constraints during training, but found the DEQ solver still diverged. We have also analyzed the instability by tracing the variation of Lipschitz constant during the adversarial training of DEQs; See Appendix E.4 for details. By comparison, the unrolling-trained DEQ requires fewer iterations in the forward solver to converge. According to the green line in Fig. 2-(b), the relative errors become lower consequently and reach 0.04 at the final state. The results coincide with [14] that the unrolling-based phantom gradient invokes implicit Jacobian regularization during training.

## 6.2 Robustness of DEQs under white-box attacks

Intriguingly, we discover the robustness accumulation effect in both the exact-trained and the unrolling-trained DEQs. We plot the highest robustness under the ready-made PGD-10 among all the intermediate states in Fig. 2-(a), with comparison to the final state robustness. It is shown that the intermediate states always exhibit much higher robustness. The accumulated robustness comes from gradient obfuscation, as the ready-made attacks fail to "directly" attack the intermediate states due to misaligned gradients. This resonates with the first challenge in Sec. 3, and similar results are observed as well in adversarially-trained neural ODEs: the large error tolerance from the ODE solvers with adaptive step sizes allows gradient masking after adversarial training [19].

The exact-trained DEQs, as we have discussed in Sec. 6.1, require larger iterations in the solver. However, it is noticed in Fig. 2-(a) that the larger the iteration is in the exact-trained DEQs, the more robust the intermediate states are under ready-made PGD-10. On the contrary, the (8/7) unrolling-trained DEQ still achieves the highest robustness at the final state. To benchmark the white-box robustness, in this section, we compare the (50/50) exact-trained DEQ-Large with the (8/7) unrolling-

Table 3: Performance (%) of the unrolling-trained DEQs under **PGD-10/AutoAttack**. The rows and the columns represent the same meanings as those in Table 2. Under the (underlined) strongest attacks, the ensemble defense achieves the best robustness performance (**in bold**).

| Arch. | Defense | Clean | Simultaneous Adjoint (PGD/AA) | | | Unrolled Intermediates (PGD/AA) | | |
|---|---|---|---|---|---|---|---|---|
| | | | Final | Intermediate | Ensemble | Final | Intermediate | Ensemble |
| Large | Final | 78.03 | 49.81/51.48 | 59.49/61.95 | 54.91/52.95 | 42.67/37.27 | 62.24/65.53 | 51.52/49.66 |
| | Early | 79.57 | 54.90/61.12 | 39.19/55.47 | 42.76/56.84 | 51.90/56.86 | 29.38/25.41 | 34.20/49.74 |
| | Ensemble | 79.67 | 51.52/56.06 | 52.43/58.69 | 49.47/55.02 | 49.02/50.45 | 55.10/58.63 | **47.12/48.37** |
| XL | Final | 82.92 | 55.80/58.21 | 55.80/58.21 | 67.30/64.24 | 48.58/43.97 | 65.94/72.76 | 58.23/69.83 |
| | Early | 80.12 | 51.40/58.92 | 51.40/58.92 | 60.78/62.98 | 52.08/56.58 | 55.70/62.67 | 48.88/62.87 |
| | Ensemble | 81.17 | 52.87/58.08 | 52.87/58.08 | 61.40/62.23 | **51.70/54.09** | 59.71/66.90 | 53.23/56.45 |

trained one [3]. We integrate the estimated intermediate gradients in Sec. 4 as different alternatives in PGD-10 for white-box evaluation. Among all the attack candidates based on intermediate gradients, we select the one that leads to the largest robustness deterioration on the early-state defense in Sec. 5.2 and report the results in Table 2. Ablation studies on the performance of the attack candidates with estimated gradients at different intermediates can be found in Sec. 6.4. We also include the report of memory usage for the defense strategies in Appendix F.1, and the running time complexity analysis for the white-box attacks in Appendix F.2.

Shown in Table 2, for the unrolling-trained DEQ, unrolling the intermediate states results in the strongest attack to the final state and early state defenses. While the robustness accuracies under final and intermediate attacks are improved and better balanced with the ensemble state defense, the ensemble attack leads to the largest performance drop in this case, arriving at the overall white-box robustness of $47.12\%$. The estimated intermediate gradients based on simultaneous adjoint process also shows significant attack performance for the exact-trained DEQ with large solver iterations. After maximizing the minimum robustness under all attacks across all defense techniques, the overall white-box robustness is $39.18\%$. In addition, all attacks significantly deteriorate the robustness of the DEQs without adversarial training, indicating that the attacks leveraged in Table 2 are reliably strong (detailed in Sec. 6.5). Considering the superior robustness of the unrolling-trained DEQ as well as its training efficiency, we proceed to experiment with unrolling-trained DEQs for further evaluation.

### 6.3 Comparison between DEQs and deep networks

In this section, we further provide a thorough evaluation by benchmarking the white-box robustness performance of the unrolling-trained DEQ-Large and DEQ-XL under both PGD-10 and AutoAttack.

Table 3 shows the robustness performance of the unrolling-trained DEQ-Large and DEQ-XL. According to the results, the gradient ensemble attacks are more effective in defeating the early-state defense than the final-state defense. The ensemble attack is the most threatening on the ensemble defense in DEQ-Large. The attack with the gradient at the final state leads to the most substantial performance drop on the ensemble defense in DEQ-XL.

In Table 3, we find that the PGD-10 attack brings more significant performance drops than AutoAttack does in many settings. The results differ from the case in the robustness of deep networks [13]. The phenomenon originates from the difference between intermediate-state attacks and alternative defense strategies. AutoAttack will overfit to the provided gradients at the intermediate or the averaged states, thus generating less threatening adversaries on the defenses based on other states. In addition, the intermediate gradients can also be inaccurate, as they only serve as approximations.

Table 4: The comparison of robustness performance (%) between the DEQs with the ensemble defense and the deep networks of similar sizes. For the DEQs, the weakest robustness under all attacks in Table 3 is reported. For the deep networks, we report the results in [29].

| Arch. | Clean | PGD-10 | AA | #Params |
|---|---|---|---|---|
| ResNet-18 | 82.52 | 53.58 | 48.51 | 10M |
| DEQ-Large | 79.67 | 47.12 | 48.37 | 10M |
| WRN-34-10 | 86.07 | 56.60 | 52.19 | 48M |
| DEQ-XL | 81.17 | 51.70 | 54.09 | 48M |

The minimum robustness under all types of attacks represents the robustness of a defense strategy. We thus take the most robust defenses for DEQ-Large and DEQ-XL, and compare them with the

---

[3]The robustness of the (8/7) exact-trained DEQ-Large is much lower than its unrolling-trained counterpart because of the violated fixed-point structure, as shown in Sec. 6.1 and Table 1

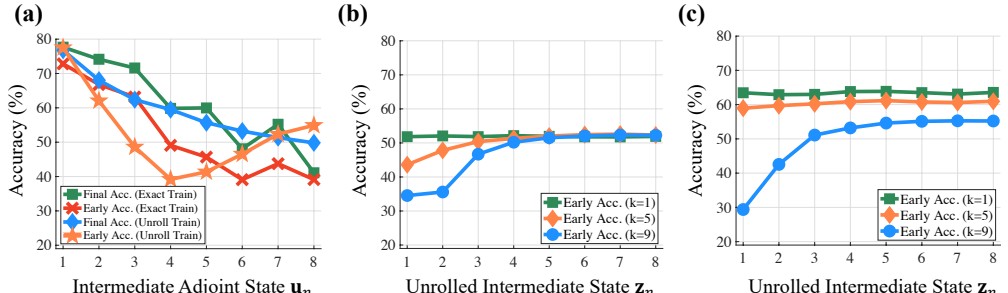

Figure 3: Ablation study on the gradient estimations at different intermediate states. (a) Different robustness performance under PGD-10 with different intermediate adjoint states $\mathbf{u}_n$ as the surrogate gradient. For the approach in Sec. 4.1, $\mathbf{u}_4$ leads to the largest robustness drop in the early state $\mathbf{z}_3$ in the unrolling-based DEQ. (b) and (c): Different unrolled intermediates $\mathbf{z}_n$ with different $k$'s in Eq. (14). the $\lambda$ in Eq. (13) is set as $0.5$ in (b) and $1$ in (c). For the method in Sec. 4.2, unrolling the state $\mathbf{z}_1$ with $k = 1$ and $\lambda = 1$ results in the largest robustness drop in the early state $\mathbf{z}_3$.

deep networks of similar parameter counts. Shown in Table 4, the empirical robustness of the DEQs is competitive with or even slightly higher than that of the ResNet-18 and WRN-34-10 models with the PGD-AT framework, respectively.

## 6.4 Ablation study on different intermediate gradients

In this section, we study the effect of the white-box attacks with gradients estimated at different intermediate states in the forward solver of the large-sized DEQs.

We first inspect the attacks with intermediate gradients acquired from each adjoint state. Fig. 3-(a) plots the robustness of the early-state and final-state defense in both the unrolling-trained and the exact-trained DEQs. For the exact-trained DEQ, due to its violated fixed-point structure, $\mathbf{u}_8$ results in the strongest attack for both the early-state and the final-state defenses. For the unrolling-trained DEQ, the estimated gradients at the consecutive adjoint states $\{\mathbf{u}_n\}$ form increasingly stronger attacks on the robustness of the final state. On the robustness at the early state ($\mathbf{z}_3$), the state $\mathbf{u}_4$ gives rise to the strongest attack, which coincides with Eq. (9) and Eq. (10) that $\mathbf{u}_{n+1}$ directly depends on $\mathbf{z}_n$.

We further explore whether the simultaneous adjoint process are aligned with the forward pass. We use each intermediate adjoint state $\mathbf{u}_n$ as the gradient surrogate in the PGD-10 attack for the unrolling-trained DEQ. Fig. 4 shows the robustness performance of all the intermediate states $\mathbf{z}_n$ in the forward pass of the unrolling-trained DEQ-Large. According to Fig. 4, it always follows that $\mathbf{u}_{n+1}$ results in the largest robustness drop of $\mathbf{z}_n$. As $\mathbf{u}_{n+1}$ directly depends on $\mathbf{z}_n$ in Eq. (10), this validates that the simultaneous adjoint process is aligned with the forward pass at each iteration in terms of adversarial robustness.

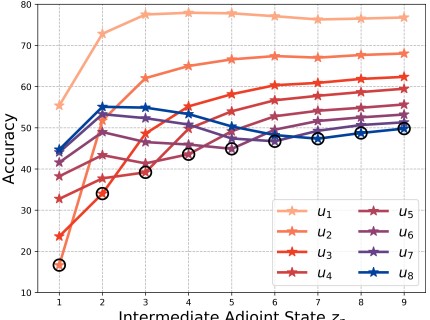

Figure 4: Alignment between the simultaneous adjoint process and the forward pass in the unrolling-trained DEQ.

We also study the effect of unrolling different intermediate states $\{\mathbf{z}_n\}$ for surrogate gradient estimation. Fig. 3-(b)/(c) illustrates the robustness of the early state $\mathbf{z}_3$ in the unrolling-trained DEQ under white-box attacks in different settings. It is shown that the number of unrolling steps for intermediate states like $\mathbf{z}_1$ and $\mathbf{z}_2$ should not be too much in order to obtain a powerful intermediate attack. The reason of this might be the inaccuracy of the unrolled intermediate gradient estimates.

The gradient estimated by unrolling $\mathbf{z}_1$ leads to the most vigorous attack on the robustness of $\mathbf{z}_3$. To understand the circumstance, we note that the unrolling-based intermediate gradient reflects only the feedback from the loss function at the unrolled state in Eq. (13) and Eq. (14). As a result, the estimated gradient may be misaligned with the unrolled state: gradients by unrolling $\mathbf{z}_3$ compose weak attack in terms of the robustness of $\mathbf{z}_3$. It is inferred that the perturbation from the unrolled intermediate gradients must still be propagated in the forward pass to induce enough threatening

Table 5: The performance (%) under the standardly-trained DEQ-Large under ready-made PGD-10.

| State | $\mathbf{z}_1$ | $\mathbf{z}_2$ | $\mathbf{z}_3$ | $\mathbf{z}_4$ | $\mathbf{z}_5$ | $\mathbf{z}_6$ | $\mathbf{z}_7$ | $\mathbf{z}_8$ |
|---|---|---|---|---|---|---|---|---|
| Clean Acc. | 38.81 | 82.62 | 89.63 | 91.77 | 92.08 | 92.29 | 92.39 | 92.53 |
| Robust Acc. | 2.00 | 0.00 | 0.00 | 0.00 | 0.00 | 0.00 | 0.00 | 0.00 |

Table 6: Performance (%) of the standardly-trained DEQ-Large [5] with all proposed adaptive attacks and defense strategies under the PGD attack. The notations for the rows and the columns are similar with those in Tables 2 and 3. Under the (underlined) strongest attacks, the ensemble defense still achieves the best robustness performance (**in bold**).

| Defense | Clean | Simultaneous Adjoint | | | Unrolled Intermediates | | |
|---|---|---|---|---|---|---|---|
| | | Final | Intermediate | Ensemble | Final | Intermediate | Ensemble |
| Final | 92.53 | 8.90 | 11.45 | 3.69 | 0.00 | 0.00 | 0.00 |
| Early | 38.81 | 6.08 | 4.54 | 3.42 | 2.00 | 2.94 | **1.31** |
| Ensemble | 87.31 | 9.12 | 6.39 | 3.48 | 0.00 | 0.00 | 0.00 |

distortion. This explains the delay that the unrolled gradient at $\mathbf{z}_1$ affects the robustness of $\mathbf{z}_3$. More ablation studies on the unrolled intermediates can be found in Appendix E.1.

## 6.5 Performance of the proposed attacks on vanilla DEQ models

In this section, we evaluate the performance of the proposed attacks on the DEQ models without adversarial training. We train a DEQ-Large on CIFAR-10 with standard training following the recipe in [5], and use the ready-made PGD-10 to attack the model. The clean accuracy of each state $\mathbf{z}_n$, as well as its robust accuracy is shown in Table 5. Different from the robustness accumulation effect in the adversarially-trained DEQs (shown in Sec. 6.2, Fig. 2-(a), and Appendix C), the ready-made PGD-10 already has a dramatic effect in attacking all the states in the standardly-trained DEQ.

We proceed to apply all the proposed attacks and defense strategies. Following Sec. 5.2, we determine the optimal timing for early exiting the standardly-trained DEQ as state $\mathbf{z}_1$. Shown in Table 6, it can be seen that all the proposed attacks can defeat the DEQ by standard training. As the white-box robustness of DEQs is assessed by the strongest defense under all attacks (minimum over all columns in a row, then maximum over the minimum of the rows), the white-box robustness of the vanilla DEQ is 1.31% with a 38.81% clean accuracy using the early-state defense. When using the final-state and the ensemble-state defense, the robustness is 0%. These results validate that all the proposed attacks are reliably strong as they all defeat the DEQ models without adversarial training.

## 7 Conclusion

We study the adversarial robustness of general DEQs, using the exact gradient and the unrolling-based phantom gradient in adversarial training for DEQs, respectively. We observe the gradient obfuscation issues in DEQs under ready-made attacks. Based on the misalignment between the forward and backward tracks, we leverage intermediate states in the forward pass to construct white-box attacks and defense strategies and benchmark the white-box robustness performance of DEQs.

While we have performed a serious comparison of white-box robustness between DEQs and deep networks, it can be seen that the performance of DEQs is on par with that of deep networks. Our empirical observations indicate that we should explore more advanced AT mechanisms for DEQs, in order to exploit their local attractor structures. A potential way is to explicitly encourage closed-loop control during training, similar to the mechanism introduced in [11]. To this end, the gradient estimation method proposed in this paper would be one of the critical ingredients for solving the misalignment between the forward/backward pass of DEQs.

## Acknowledgements

This work was supported by the National Natural Science Foundation of China (No.61925601) and Beijing Academy of Artificial Intelligence (BAAI). We appreciate all of the anonymous reviewers for their comments and suggestions on this work.

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
