gradient used in the attack for checkpoint selection is the same as the gradient used for training. The settings on the basics of DEQ training are largely followed from [5]. We leave the integration of the estimated intermediate gradients for adversary generation into AT for DEQs as future work. We use NVIDIA-3090 GPUs for all of our experiments.

Table 7: Detailed hyperparameter settings.

| Category | Settings | DEQ-Large | DEQ-XL |
|---|---|---|---|
| Architecture | Input Image Size | $32 \times 32$ | |
| | Number of Scales | 4 | |
| | # of Head Channels for Each Scale | [14, 28, 56, 112] | [20, 40, 80, 160] |
| | # of Channels for Each Scale | [32, 64, 128, 256] | [72, 144, 288, 576] |
| | Channel Size of Final Layer | 1,680 | 1,800 |
| | Activation Function | ReLU | |
| | # of Parameters | 10M | 48M |
| DEQ Solver | # of Forward Solver Iterations | 8 | |
| | # of Backward Solver Iterations | 7 | |
| | Algo. for Forward Solvers | Broyden's Method | |
| | Algo. for Backward Solvers | Broyden's Method | |
| Optimization | Optimizer | Adam | |
| | Learning Rate Schedule | cosine decay | |
| | Decay Factor | 0.1 | |
| | Epochs for Decay | [30, 60, 90] | |
| | Initial Learning Rate | 0.001 | |
| | Nesterov Momentum | 0.98 | |
| | Weight Decay | - | |
| Adv. Training | Batch Size | 96 | |
| | Training Epochs | 150 | |
| | Pretraining Steps | 16,000 | |
| | Weight of JR During Pretraining | - | |
| | Weight of JR During DEQ Training | 0.4 | |
| | Stop Epoch for JR | 90 | |
| | Unrolling-Trained: Steps $k$ | 5 | |
| | Unrolling-Trained: Damping Factor $\lambda$ | 0.5 | |
| | AT for Pretraining | No | |
| | Attack in AT for DEQ Training | ready-made PGD-10 | |
| | Grad. for Adv. Generation in AT | same with the gradient used in training | |
| | Label Smoothing | - | |

# B On the convergence of simultaneous adjoint

We start with the assumption on nonsingularity of the Jacobian inverse of $g_\theta(\mathbf{z}_n; x) = f_\theta(\mathbf{z}_n; x) - \mathbf{z}_n$:

**Assumption B.1.** For $\forall \mathbf{z} \in \mathbb{R}^d$ and $\forall \mathbf{x} \in \mathbb{R}^l$, we assume the nonsingularity of Jacobian inverse of $g_\theta(\mathbf{z}; x) = f_\theta(\mathbf{z}; x) - \mathbf{z}$. Namely,

$$\left( \frac{\partial g_\theta(\mathbf{z}; x)}{\partial \mathbf{z}} \right)^{-1} = \left( \frac{\partial f_\theta(\mathbf{z}; \mathbf{x})}{\partial \mathbf{z}} - I \right)^{-1} \tag{18}$$

exists.

We also make assumptions on the precision of $B_n$, which is the approximation of the Jacobian inverse of $g_\theta(\mathbf{z}_n; x)$.

**Assumption B.2.** Define

$$\bar{B}_n \left( \frac{\partial g_\theta(\mathbf{z}_n; x)}{\mathbf{z}_n} \right) = I - \boldsymbol{\epsilon}_n(\mathbf{x}). \tag{19}$$

We assume that $\|\boldsymbol{\epsilon}_n(\mathbf{x})\| \le 1 - \epsilon < 1$ with $\epsilon > 0$ for all $\mathbf{x}$.

With these two assumptions, with $0 < \beta < 1$, we have:

$$\|\mathbf{u}_{n+1} - \mathbf{u}^*\| \tag{20}$$

$$= \|\mathbf{u}_n - \beta \bar{B}_n \mathbf{v}_n - \mathbf{u}^*\| \tag{21}$$

$$= \left\| \mathbf{u}_n - \beta \bar{B}_n \left( \left( \frac{\partial f_\theta(\mathbf{z}_n; \mathbf{x})}{\partial \mathbf{z}} \right) \mathbf{u}_n + \frac{\partial L(\mathbf{z}_n, y)}{\partial \mathbf{z}} - \mathbf{u}_n \right) - \mathbf{u}^* \right\| \tag{22}$$

$$= \left\| \mathbf{u}_n - \beta \bar{B}_n \left( \frac{\partial g_\theta(\mathbf{z}_n; x)}{\partial \mathbf{z}} \right) \mathbf{u}_n - \beta \bar{B}_n \frac{\partial L(\mathbf{z}_n, y)}{\partial \mathbf{z}} - \mathbf{u}^* \right\| \tag{23}$$

$$= \left\| \mathbf{u}_n - \beta \left( I - \boldsymbol{\epsilon}_n(\mathbf{x}) \right) \mathbf{u}_n - \beta \bar{B}_n \frac{\partial L(\mathbf{z}_n, y)}{\partial \mathbf{z}} - \mathbf{u}^* \right\| \tag{24}$$

$$= \left\| \left( (1 - \beta)I + \beta \boldsymbol{\epsilon}_n(\mathbf{x}) \right) \mathbf{u}_n + \beta \left( I - \boldsymbol{\epsilon}_n(\mathbf{x}) \right) \left( I - \frac{\partial f_\theta(\mathbf{z}_n; \mathbf{x})}{\partial \mathbf{z}} \right)^{-1} \frac{\partial L(\mathbf{z}_n, y)}{\partial \mathbf{z}} - \mathbf{u}^* \right\| \tag{25}$$

$$= \left\| \left( (1 - \beta)I + \beta \boldsymbol{\epsilon}_n(\mathbf{x}) \right) (\mathbf{u}_n - \mathbf{u}^*) + \beta \left( I - \boldsymbol{\epsilon}_n(\mathbf{x}) \right) \left( \left( I - \frac{\partial f_\theta(\mathbf{z}_n; \mathbf{x})}{\partial \mathbf{z}} \right)^{-1} \frac{\partial L(\mathbf{z}_n, y)}{\partial \mathbf{z}} - \mathbf{u}^* \right) \right\| \tag{26}$$

$$\le \|(1 - \beta)I + \beta \boldsymbol{\epsilon}_n(\mathbf{x})\| \|\mathbf{u}_n - \mathbf{u}^*\| + \beta \|I - \boldsymbol{\epsilon}_n(\mathbf{x})\| \left\| \left( I - \frac{\partial f_\theta(\mathbf{z}_n; \mathbf{x})}{\partial \mathbf{z}} \right)^{-1} \frac{\partial L(\mathbf{z}_n, y)}{\partial \mathbf{z}} - \mathbf{u}^* \right\|. \tag{27}$$

Define

$$\mathbf{w}_n = \left( I - \frac{\partial f_\theta(\mathbf{z}_n; \mathbf{x})}{\partial \mathbf{z}} \right)^{-1} \frac{\partial L(\mathbf{z}_n, y)}{\partial \mathbf{z}}. \tag{28}$$

According to the definition of $\mathbf{u}^*$ in Eq. (2), it follows that $\mathbf{w}_n \to \mathbf{u}^*$ when $\mathbf{z}_n \to \mathbf{z}^*$. $\{\mathbf{w}_n\}$ has the highest convergence rate to $\mathbf{u}^*$ as it is the most precised estimation of $\mathbf{u}^*$ at step $n$. According to Assumption B.2 and $0 < \beta < 1$, the first term in Eq. (27) follows

$$\|(1 - \beta)I + \beta \boldsymbol{\epsilon}_n(\mathbf{x})\| \le (1 - \beta) + \beta \|\boldsymbol{\epsilon}_n(\mathbf{x})\| \le 1 - \beta \epsilon < 1. \tag{29}$$

Substituting Eq. (29) and the upper bound $\epsilon_0$ into Eq. (27), we have

$$\|\mathbf{u}_{n+1} - \mathbf{u}^*\| \le (1 - \beta \epsilon) \|\mathbf{u}_n - \mathbf{u}^*\| + \|\mathbf{w}_n - \mathbf{u}^*\|, \tag{30}$$

thus

$$\frac{\|\mathbf{u}_{n+1} - \mathbf{u}^*\|}{\|\mathbf{u}_n - \mathbf{u}^*\|} \le 1 - \beta \epsilon + \frac{\|\mathbf{w}_n - \mathbf{u}^*\|}{\|\mathbf{u}_n - \mathbf{u}^*\|}. \tag{31}$$

We omit the second term in the right-hand side of (31) under the mild assumption of the strictly higher convergence rate of $\{\mathbf{w}_n\}$. As a result, when $n > N$ with $N$ sufficiently large, it follows that $\{\mathbf{u}_n\}$ converges to $\mathbf{u}^*$ as $\|\mathbf{u}_{n+1} - \mathbf{u}^*\| < \|\mathbf{u}_n - \mathbf{u}^*\|$. However, we do *not* require the convergence of $\mathbf{u}_n$ as we only use them in Eq. (11) as gradient *estimations* to construct *intermediate attacks*. In practice, we tune $\beta$ to facilitate the strongest attacks. For the (8/7) unrolling-trained DEQ-Large/XL, we set $\beta = 0.5$; for the (50/50) exact-trained DEQ-Large, we set $\beta = 0.05$.

## C  The robustness accumulation effect

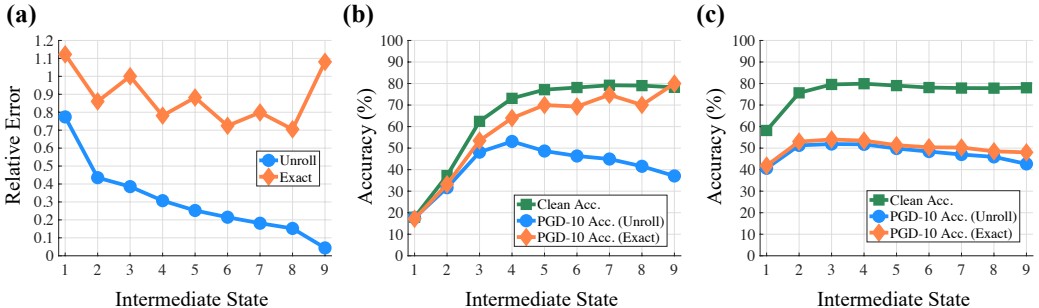

Figure 5: The performance of all intermediate states $\mathbf{z}_n$ in the exact-trained and the unrolling-trained DEQ-Large with 8 forward and 7 backward iterations. (a) Relative error at each intermediate state $\mathbf{z}_n$. The fixed-point structure in the exact-trained DEQ is violated as all relative errors are higher than 1.0. (b) and (c): Robustness evaluated at different intermediate states in (b) the exact-trained and (c) the unrolling-trained DEQs. The unrolling-based phantom gradient forms stronger adversaries in the ready-made PGD-10 attack. The intermediate state $\mathbf{z}_3$ exhibits the strongest robustness in the unrolling-trained DEQ, while $\mathbf{z}_4$ exhibits the strongest robustness in the exact-trained DEQ.

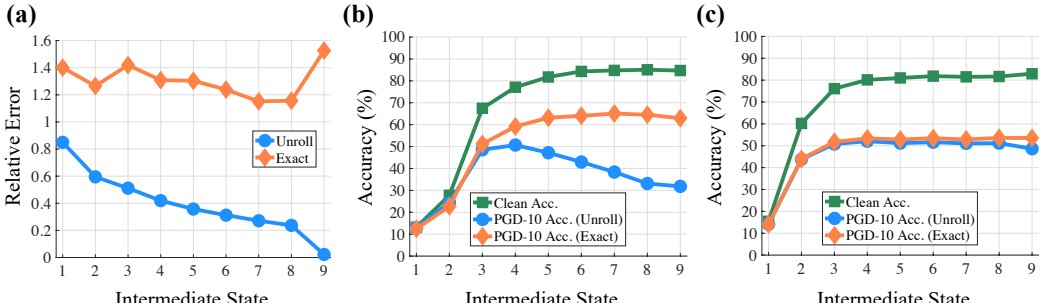

Figure 6: The performance of all intermediate states $\mathbf{z}_n$ in the exact-trained and the unrolling-trained DEQ-XL with 8 forward and 7 backward iterations. (a) Relative error at each intermediate state $\mathbf{z}_n$. The fixed-point structure in the exact-trained DEQ is violated as all relative errors are higher than 1.0. (b) and (c): Robustness evaluated at different intermediate states in (b) the exact-trained and (c) the unrolling-trained DEQs. The unrolling-based phantom gradient forms stronger adversaries in the ready-made PGD-10 attack. For both DEQs, the intermediate state $\mathbf{z}_4$ exhibits stronger robustness than other states.

Figures 5 and 6 the robustness accumulation effect in the DEQs with 8 forward and 7 backward iterations. For both the exact-trained and the unrolling-trained DEQs, the robustness accumulation effect exists because the black-box solvers have obfuscated the gradients used in the ready-made attacks. It can also be seen that the fixed-point structures are broken in the exact-trained DEQ-Large and DEQ-XL under this setting. Figure 2 show that the exact-trained DEQs with more iterations in the solvers can retain the fixed-point structure (more details in Appendix E.3).

# D Comparison with the adjoint Broyden method

Similar to our simultaneous adjoint design, concurrent work [33] propose the adjoint Broyden method to share the approximated Jacobian inverse $B_n$ into the backward pass. Their motivation, however, is to accelerate DEQ training while we integrate the simultaneous adjoint into ready-made attacks to facilitate white-box robustness evaluation. In this section, we compare the adjoint Broyden method with our simultaneous adjoint in terms of the effect of intermediate/ensemble attacks.

Table 8: Comparison between the proposed simultaneous adjoint and the adjoint Broyden method.

| Arch. | Defense | Clean | PGD : Simultaneous Adjoint / Adjoint Broyden | | |
|-------|---------|-------|-------|-------|-------|
| | | | Final | Intermediate | Ensemble |
| DEQ-Large | Final | 78.03 | **49.81**/52.34 | 59.49**/58.22** | **54.91**/58.06 |
| | Early | 79.57 | 54.90/**45.29** | **39.19**/43.48 | **42.76**/45.47 |
| | Ensemble | 79.67 | 51.52/**49.24** | 52.43/**49.24** | **49.47**/53.04 |
| DEQ-XL | Final | 82.92 | **55.80**/65.00 | **55.80**/65.00 | **67.30**/71.20 |
| | Early | 80.12 | **51.40**/59.26 | **51.40**/59.26 | **60.78**/65.67 |
| | Ensemble | 81.17 | **52.87**/60.22 | **52.87**/60.22 | **61.40**/66.69 |

Table 8 shows the comparison between our simultaneous adjoint and the adjoint Broyden method with PGD as the attack. The proposed simultaneous adjoint results in stronger white-box attacks in general. The adjoint Broyden method also yields better performance in some cases. We further compare among the proposed simultaneous adjoint, the adjoint Broyden method, and the unrolled intermediates in Table 9.

Table 9: Comparison among the proposed simultaneous adjoint, the adjoint Broyden method, and the (proposed) unrolled intermediates with PGD.

| Arch. | Defense | Clean | PGD : Simultaneous Adjoint / Adjoint Broyden / Unrolling Intermediates | | |
|-------|---------|-------|-------|-------|-------|
| | | | Final | Intermediate | Ensemble |
| DEQ-Large | Final | 78.03 | 49.81/52.34/**42.67** | 59.49/**58.22**/62.24 | 54.91/58.06/**51.52** |
| | Early | 79.57 | 54.90/**45.29**/51.90 | 39.19/43.48/**29.38** | 42.76/45.47/**34.20** |
| | Ensemble | 79.67 | 51.52/49.24/**49.02** | 52.43/**49.24**/55.10 | 49.47/53.04/**47.12** |
| DEQ-XL | Final | 82.92 | 55.80/65.00/**48.58** | **55.80**/65.00/65.94 | 67.30/71.20/**58.23** |
| | Early | 80.12 | **51.40**/59.26/52.08 | **51.40**/59.26/55.70 | 60.78/65.67/**48.88** |
| | Ensemble | 81.17 | 52.87/60.22/**51.70** | **52.87**/60.22/59.71 | 61.40/66.69/**56.45** |

Table 9 shows that the unrolled intermediates still have the dominant effect in facilitating white-box attacks in general. In the future work, we will integrate our simultaneous adjoint/the adjoint Broyden method into the adversarial training process of DEQs and validate their white-box robustness.

# E Ablation studies

## E.1 Settings of the unrolled intermediates

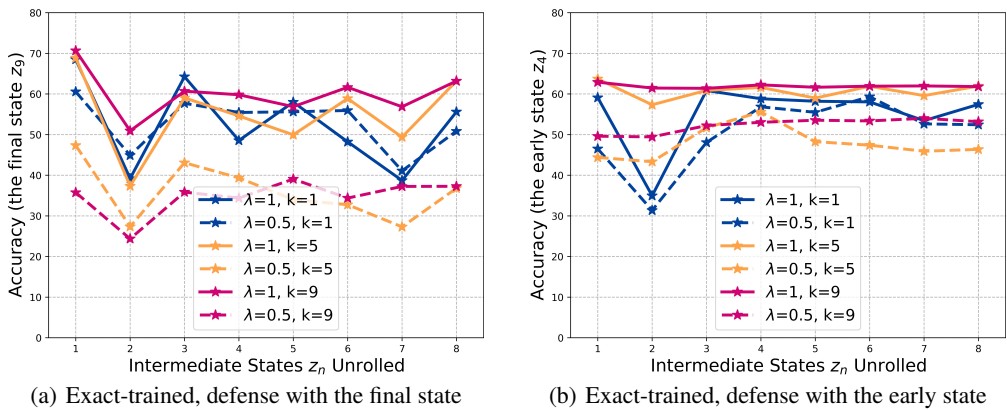

(a) Exact-trained, defense with the final state  (b) Exact-trained, defense with the early state

Figure 7: Ablation on unrolling different intermediate states with different steps in exact-trained DEQ-Large.

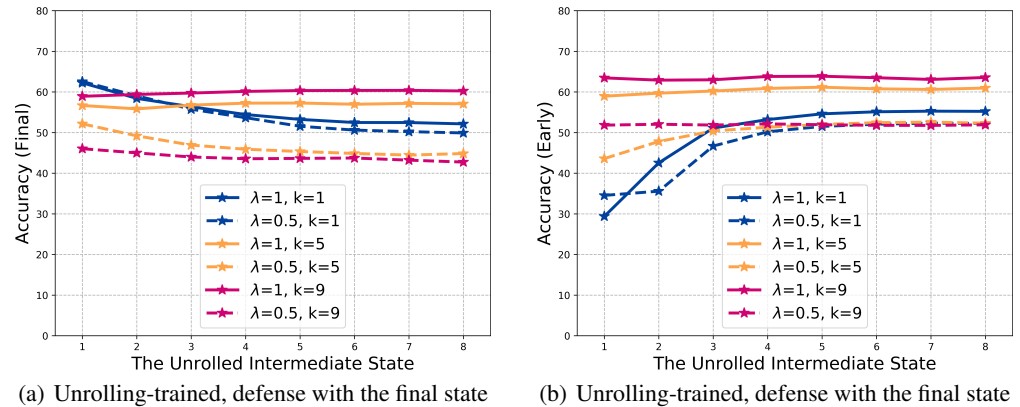

(a) Unrolling-trained, defense with the final state  (b) Unrolling-trained, defense with the final state

Figure 8: Ablation on unrolling different intermediate states with different steps in unrolling-trained DEQ-Large.

## E.2 The backward solver iteration threshold in attacks with exact gradients

The exact gradient in Eq. (2) is solved by an independent fixed-point iteration process solely based on $\mathbf{z}^*$. While this iteration process is not aligned with the forward pass, one might still question whether the intermediate states in the backward solver can lead to strong attacks.

We address the question by experimenting with the unrolling-trained DEQ-Large. We use the exact gradient solved by a backward solver for adversary generation in PGD-10. While the default iteration threshold is 7, we investigate the capability of all the 7 intermediate states in constructing adversaries. We also increase the number of the iterations up to 20 to see whether the robustness is degraded. The results are shown in Fig. 9.

From Fig. 9, it can be seen that the gradients from less iterations result in less powerful attacks for both the final and the early states. With increased backward iterations, the robustness of both states is dropped by about 1%. However, as shown in Table 2, PGD-10 with the unrolling-based phantom gradient results in the robustness of 42.67 for the final state and 51.90 for the early state, both of which lower than those in Fig. 9. It is thus concluded that alternating the backward iterations in the exact gradient does not have significant effect on deteriorating the robustness of the unrolling-trained DEQ.

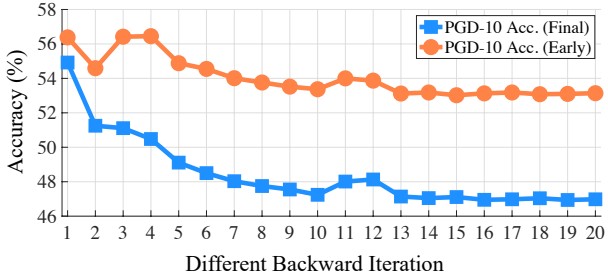

Figure 9: Robustness of the final state as well as the early state in the unrolling-trained DEQ-Large under PGD-10. The gradients in the attacks are returned from the backward solver with different iterations.

### E.3 Solver iteration thresholds in the exact-trained DEQs

We increase the iteration thresholds of the forward and the backward solvers in the exact-trained DEQs to stabilize adversarial training. We experiment with the exact-trained DEQ-Large and report the results in Table 10 (which is also illustrated in Fig. 2-(a)).

Table 10: Exact-trained DEQ-Large with larger iteration thresholds in the forward and the backward solvers during training. The "PGD, Unroll" column, demonstrating the accuracy of the final and the top intermediate state under ready-made PGD attacks with the unrolling-based phantom gradient, is used to illustrate Fig. 2-(a).

| Training | (ForIter./BackIter) | Defense | Clean | PGD, Exact | PGD, Unroll | Speed (Samples/s) | Note |
|---|---|---|---|---|---|---|---|
| Exact | (8/7) | Final | 78.24 | 79.97 | 37.07 | 28.3 | Grad. Obfus. |
| | | Early | 73.12 | 63.95 | 53.09 | | |
| | (18/20) | Final | 73.16 | 39.92 | 40.00 | 22.3 | - |
| | | Early | 72.02 | 51.44 | 51.07 | | |
| | (30/30) | Final | 81.17 | 41.51 | 37.12 | 7.6 | - |
| | | Early | 87.72 | 65.22 | 62.74 | | |
| | (50/50) | Final | 73.51 | 37.29 | 36.70 | 5.3 | - |
| | | Early | 86.98 | 74.04 | 73.93 | | |
| Unrolling | (8/7) | Final | 78.03 | 48.03 | 42.67 | **35.4** | - |
| | | Early | 79.57 | 54.01 | 51.90 | | |

From Table 10, it is witnessed that increasing the forward and the backward iterations imposes stabilization effect on DEQ training with the exact gradient. As a consequence, no false-positive robustness is observed in the settings of iteration pairs $(18/20)$. The relative error for each state $\mathbf{z}_n$ in the forward pass under the $(18/20)$ setting is shown in Fig. 10-(a).

In this setting, the forward solver may require less number of iterations than the threshold 18. Here we plot the relative errors of the first 11 states in the forward solver. From Fig. 10-(a) (also the blue line in Fig. 2-(b)), the $(18/20)$ exact-trained DEQ does not violate its fixed-point structure, thus the gradient obfuscation issue is avoided. However, both of its training speed and robustness performance are outperformed by the unrolling-trained DEQ in the $(8/7)$ setting.

Similar to Figures 5 and 6, we also compare the robustness accumulation effect between the $(18/20)$ exact-trained DEQ with the $(8/7)$ unrolling-trained DEQ. Figure 10-(b) shows the comparison: clean and PGD-10 accuracies of both the final and the top intermediate states in the (18/20) exact-trained DEQ are lower than the (8/7) unrolling-trained DEQ. This is similar with the conclusion drawn from Table 2.

### E.4 Jacobian regularization weights during training

Another way to stabilize the adversarial training process is to impose stricter regularization on the DEQs. [5] propose Jacobian regularization to stabilize the standard training of DEQs. In this section, we vary the weight $\gamma$ of the Jacobian regularization during DEQ training to its effect. In standard

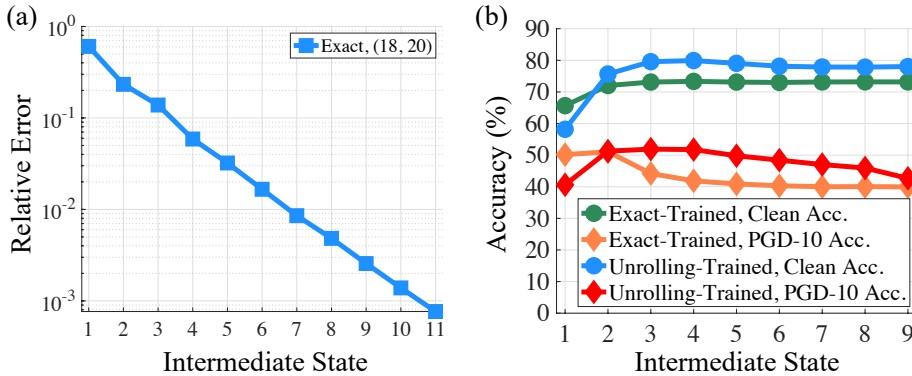

Figure 10: Performance of the (18/20) exact-trained DEQ-Large. (a) The relative error of each state $\mathbf{z}_n$ in the forward pass of the exact-trained DEQ-Large under the $(18/20)$ setting. (b) Robustness accumulation effect in DEQ-Large: the (18/20) exact-trained DEQ v.s. the (8/7) exact-trained DEQ.

training, the weight is set as $0.4$. In this section, we sweep $\gamma$ over $\{0.4, 0.8, 1.2, 1.6, 2.0, 3.2\}$ to train DEQ-Large with the exact gradient with 8 forward and 7 backward iterations (8/7).

Similar to Fig. 2-(b), we plot the relative error at each $\mathbf{z}_n$'s under different $\gamma$'s settings.

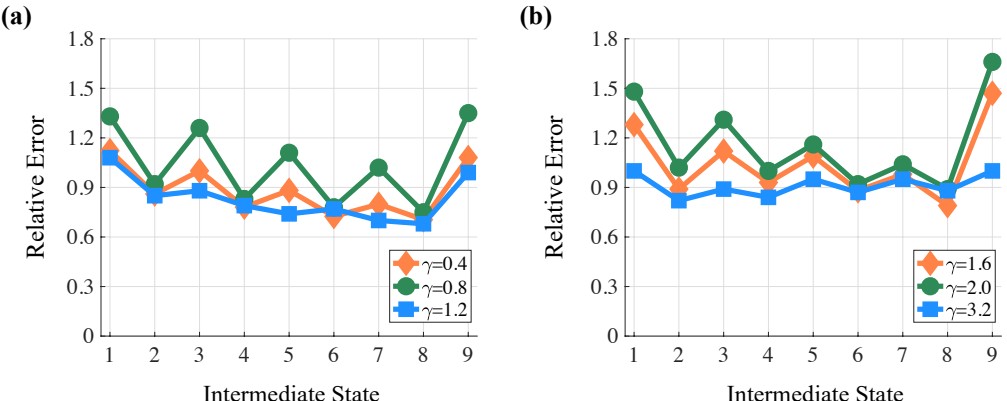

Figure 11: Relative errors at each intermediate state $\mathbf{z}_n$ under different $\gamma$'s settings.

Illustrated in Fig. 11, all of the relative errors are larger than $0.6$, indicating the violation of the fixed-point structure. We have also explored the stability of the adversarial training process in the $\gamma = 3.2$, (8/7) exact-trained setting and the $\gamma = 0.4$, (8/7) unrolling-trained setting. We calculate the averaged spectral radius on the development set for all the checkpoints along the training, and plot them, together with their accuracy under the ready-made PGD-10 attack, in Figs 12 and 13.

Shown in Fig. 13, the $\gamma = 0.4$, (8/7) unrolling-trained DEQ-Large model always has a spectral radius less than or around $1.0$. In contrast, the spectral radius of the $\gamma = 3.2$, (8/7) exact-trained DEQ-Large model becomes far larger than $1.0$ since the 65-th checkpoint in Fig. 12. This coincides with the violated fixed-point structure, although achieving high ($> 60$, but false-positive) accuracy under the ready-made PGD-10 attack. In this work, we use the checkpoint with the highest robustness under the ready-made PGD10 along the adversarial training process for our study. We leave the study of the white-box robustness evaluation for other checkpoints in future work.

When the fixed-point structure is broken, query-based attacks have a drastic effect on reducing the classification accuracy of DEQs (see Table 1). We use SQUARE to attack these exact-trained DEQ-Large models with varied Jacobian regularization weight $\gamma$'s.

Table 11 compares the performance of the models with different $\gamma$'s under SQUARE and the ready-made PGD attack. For the (8/7) exact-trained DEQ-Large with varied $\gamma$'s, the SQUARE attack

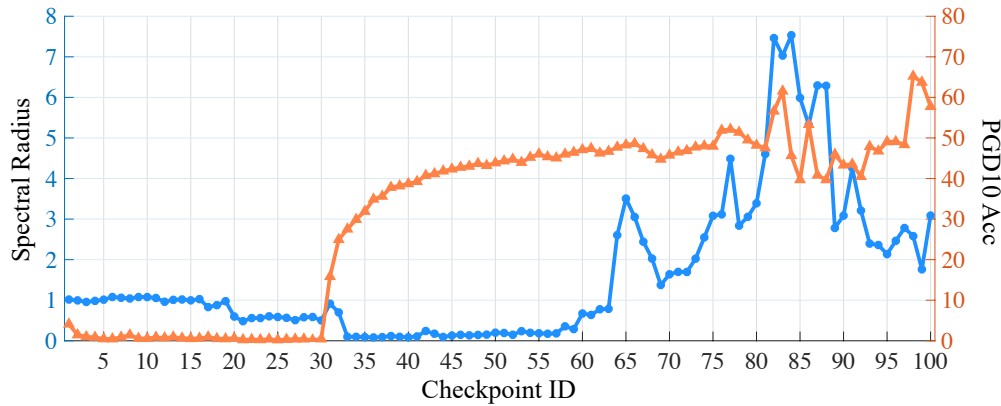

Figure 12: $\gamma = 3.2$, (8/7) exact-trained DEQ-Large. The trace of spectral radius on the development set and the accuracy of each checkpoint under the ready-made PGD-10 attack. The blue line traces the spectral radius, and the orange line traces the accuracy under ready-made PGD-10.

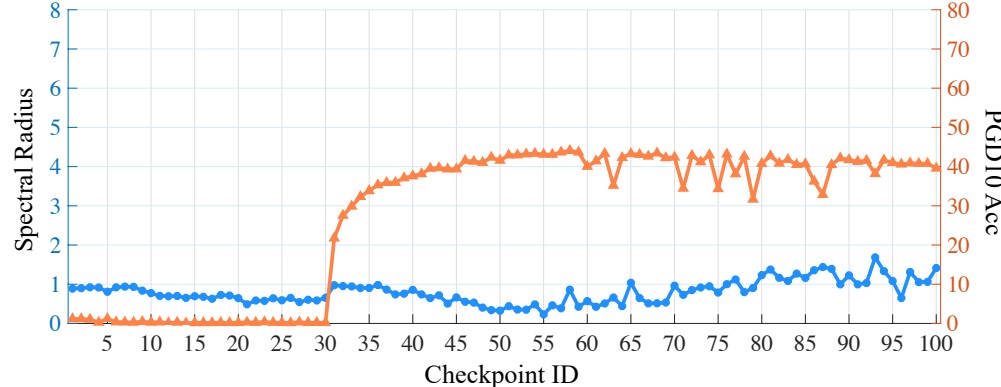

Figure 13: $\gamma = 0.4$, (8/7) unrolling-trained DEQ-Large. The trace of spectral radius on the development set and the accuracy of each checkpoint under the ready-made PGD-10 attack. The blue line traces the spectral radius, and the orange line traces the accuracy under ready-made PGD-10.

appears to be more powerful than the ready-made (gradient-based) PGD attack and always leads to severe robustness degradation. According to [2, 8], such a phenomenon indicates gradient obfuscation. For the exact-trained DEQ-Large with more iterations in the DEQ solver and the unrolling-trained DEQ-Large, as they have retained the fixed-point structure (Fig. 2-(b) and Fig. 10-(a)), we find the gradient-based PGD attack to be more effective. However, we emphasize that the two models *still* suffer from gradient obfuscation: shown in Fig. 10-(b), robustness accumulation effect is observed in both of the models (despite the retained fixed-point structure). The reason of the effect, as we have mentioned in Sec. 6.2, is that the black-box solver in DEQs results in misaligned gradients, which avoids the ready-made attacks to "directly" attack the intermediate states. This has motivated us to propose intermediate/ensemble attacks and defenses for white-box robustness evaluation.

To summarize, we have studied two types of gradient obfuscation in our work (see Table 12). The violated fixed-point structure can be remedied by different techniques: we adopt unrolling-trained DEQs in Sec. 6.1, increase the solver iterations in Sec. 6.2 and Appendix E.3, and attempt with stricter regularization in Appendix E.4. We will explore with more regularization techniques in future work. However, the robustness accumulation effect *always* exists. This ultimately urges the necessity of white-box robustness evaluation, and we propose several white-box attacks and defenses in this work. The two types of gradient obfuscation also echo with the two challenges in Sec. 3.

Table 11: (8/7) exact-trained DEQ-Large with varied Jacobian regularization weight $\gamma$'s, in comparison with ($\gamma = 0.4$): the (18/20) exact-trained DEQ-Large and the (8/7) unrolling-trained DEQ-Large.

| $\gamma$ | 0.4 | 0.8 | 1.2 | 1.6 | 2.0 | 3.2 | 0.4 (18/20) | 0.4 (Unroll) |
|---|---|---|---|---|---|---|---|---|
| Clean | 78.24 | 82.27 | 80.64 | 78.82 | 67.34 | 66.70 | 73.16 | 78.03 |
| PGD (*ready-made*) | 79.97 | 71.00 | 48.71 | 62.23 | 61.77 | 65.22 | 39.92 | 42.67 |
| SQUARE | 5.95 | 10.00 | 32.54 | 23.85 | 4.21 | 2.72 | 47.20 | 45.34 |

Table 12: To summarize our work: the two types of gradient obfuscation that we have studied, and what we have done for our white-box robustness evaluation.

| State investigated | Phenomenon observed | The reason of the phenomenon | What we have done |
|---|---|---|---|
| Final | SQUARE is more effective | Violated fixed-point structure | "Fix" the structure by: Using the unrolling-trained DEQs; Increasing the iterations in the solver; Attempting with varied regularization weights. |
| Intermediate | Robustness accumulation | Black-box solvers | Propose several white-box attacks and defenses. |

# F  Memory and time complexity

## F.1  On the O(1) memory concern of the proposed attacks and defenses

The unique property of DEQs lies in its O(1) memory consumption. It is therefore necessary to study whether the O(1) memory constraint still applies in the proposed attacks and defenses.

From the attacking aspect, white-box attackers are assumed to have full access to the model, therefore they are allowed to "open the black box" to trace all the intermediate steps in the solver. To fully evaluate the worst-case performance of models, white-box attackers are usually not constrained by the O(1) memory. From the defending aspect, as stated in Sec. 5.2, our defense methods still require only O(1) memory. For early state defense, we determine the optimal time to early exit the solver on the development set offline for once and then fix the early exit step during testing. For ensemble state defense, we maintain an accumulator ($\text{ret} += \mathbf{z}_n$) without storing the intermediate states of $\mathbf{z}_n$ and output with $\text{ret}/N$ instead of $\mathbf{z}_N$: Table 13 shows the empirical results on memory usage.

Table 13: The memory usage of different defense strategies used in the (8/7) unrolling-trained DEQ-Large. No extra computation is needed.

| Defense | Mem (GB) |
|---|---|
| Final | 3.77 |
| Early | 3.77 |
| Ensemble | 3.77 |

## F.2  On the time complexity concern of simultaneous adjoint

The core idea of the simultaneous adjoint is to **reuse** the approximated Jacobian inverse $B_n$ in the forward calculation of $\mathbf{z}_n$ when calculating the adjoint state $\mathbf{u}_n$. As a result, the "approximated Jacobian inverse" in the simultaneous adjoint does not need extra calculation. This is different from the original DEQ design where the forward and the backward passes are decoupled by separate fixed-point solvers, where different $B_n$'s need to be maintained separately.

Specifically, compared with the original forward pass (Eqs. (7) and (8)), the simultaneous adjoint calculation augments it with Eqs. (9) and (10). The time complexity of Eq.(10) is equilavent to Eq.(7). Eq. (9) calculates the residual of Eq.(3), the fixed-point equation of the backward pass, at $\mathbf{u}^* = \mathbf{u}_n$ and $\mathbf{z}^* = \mathbf{z}_n$. The complexity of this calculation is equivalent to just the **residual evaluation** when solving for the exact gradient in Eq.(3).

In practice, the running time for the adaptive PGD-10 attack with the final adjoint state in "Simultaneous Adjoint" is 6,479ms per batch. In comparison, the running time for the adaptive PGD-10 attack with the unrolled final state in "Unrolled Intermediates" is 5,369ms per batch. It can be seen that the introduced computational burden of Eqs.(9) and (10) does not take the majority of the running time.

# G    Limitations and Broader Impact

Adversarial attacks are fatal to deep learning methods, causing severe security risks in model deployment. To this end, reliable techniques are needed to defend against the attacks. In this work, we study the white-box robustness of general DEQ models. We observe the gradient obfuscation effect with ready-made attacks and propose several white-box attacks and defenses to facilitate white-box robustness evaluation. Our work contributes to the safety of general DEQs in white-box settings.

In this work, we did not test our methods on the adversarially-trained DEQs on ImageNet due to the time limit. Recent work [4, 5] has shown that DEQ models work well on large-scale vision tasks, including ImageNet and Cityscapes. Given this, we could apply advanced adversarial training algorithms [41] to DEQs, which preserves the scalability of our methods. We also leave the white-box robustness evaluations on ImageNet as our future work. In addition, future work also includes efficient stabilization of the adversarial training procedure, as well as advanced white-box attacks and defense strategies for DEQs.