# OpenReview forum: "A Closer Look at the Adversarial Robustness of Deep Equilibrium Models"
_NeurIPS.cc/2022/Conference — NeurIPS 2022 Accept_

### Official Review · Reviewer_2UMt · 2022-06-24

**Rating:** 6
**Confidence:** 3
**Soundness:** 3 good
**Presentation:** 3 good
**Contribution:** 2 fair

**Summary:**

This paper studies the robustness issue in deep equilibrium models.
Specifically, the paper points out the misalignment between forward and backward tracks of DEQ due to the black-box solver.
To address this issue, the paper proposes two gradient estimation methods:
an adjoint process and unrolling the intermediate states.
Those two proposed methods allow accurate robustness measurement and effective adversarial training.

**Questions:**

Could the author elaborate on the advantages of using adversarially trained DEQs models over the traditional deep convolutional networks?

**Limitations:**

The limitations of this proposed method are discussed in the conclusion.
The stability of adversarial training is indeed one concern of DEQ models.
Meanwhile, I think showing the advantages of using robust DEQs over deep CNNs is also not very clear at this point.

**Strengths And Weaknesses:**

Strength:
The observation of misalignment between forward and backward passes in DEQs is a significant contribution to the adversarial robustness research.
The proposed two gradient estimation methods are helpful for accurately evaluating the robustness of DEQs.

Weakness:
Adversarially trained DEQ models do not show significant advantages compared with conventional deep neural networks, as shown in Table 3. Their performances are comparable.

---

> ### Author Response · Authors · 2022-08-02
> **Thank you for the supportive review**
>
> Thank you for the supportive review and kind suggestions. We have uploaded a revision of our paper.
>
> ***Question 1: The adversarially-trained DEQs do not significantly outperform the traditional deep neural networks in terms of robustness***
>
> The motivation of our paper is to comprehensively evaluate the adversarial robustness of DEQs. Intuitively, the fixed-point structure in DEQs can be viewed as a local attractor, which is expected to be more stable to small input perturbations compared to deep networks. However, as observed in our experiments, directly applying existing adversarial training frameworks (e.g., PGD-AT) on DEQs does not show significant advantages compared to conventional deep networks.
>
>
> Our empirical observations indicate that we should explore more advanced AT mechanisms for DEQs, in order to exploit their local attractor structures (similar to the feedback regulation in control theory). A potential way is to explicitly encourage closed-loop control during training, similar to the mechanism introduced in [1]. To this end, the gradient estimation method proposed in this paper would be one of the critical ingredients for solving the misalignment between the forward/backward pass of DEQs.
>
> ***Reference:***
>
> [1] Towards Robust Neural Networks via Close-loop Control. ICLR 2021

---

### Official Review · Reviewer_LJfG · 2022-07-01

**Rating:** 5
**Confidence:** 3
**Soundness:** 4 excellent
**Presentation:** 2 fair
**Contribution:** 2 fair

**Summary:**

This paper investigates the adversarial robustness of DEQs. In order to apply the standard attack algorithms (e.g. PGD attacks), the authors propose to estimate the gradient of intermediate states with two different approaches. The authors also propose techniques to improve the robustness of DEQs.

**Questions:**

* How do you apply the two different intermediate gradient estimation techniques in Section 4.1&4.2 to your experiments? How do they relate with the previous two training approaches "exact" and "unrolling"?

* What is the performance of your proposed attack against vanilla-trained DEQs? Which attack will have the better performance, regardless of how the model is trained (e.g. exact/unrolling)?

**Limitations:**

I do not see the discussion of the limitation in the paper. As far as I can tell, I think the efficiency of adv-DEQs would be a concern compared with standard DL models. In addition, I doubt whether the approach can be applied to larger-scale tasks such as ImageNet.

**Strengths And Weaknesses:**

## Strengths

* The paper proposes and investigates an interesting topic - the empirical robustness of DEQs. The authors identify the key problem of empirical adv attacks on general DEQs and propose methods for both attacks and defenses.

* As far as I can tell, the proposed approach is correct and intuitively makes sense. The authors propose two techniques to approximate the gradient of intermediate states, based on which they can perform the adversarial attack and adv training algorithms. These proposed approaches are clear and correct.

## Weaknesses

* The attack on vanilla-trained DEQs are missing in the evaluation, which is a key part of the story. To me, the logic of the paper would be 1) authors propose an adv attack against DEQs; 2) authors utilize the attack to do PGD adversarial training. Although the authors show that the adv-trained models indeed have a good performance against the proposed attacks, they do not show that the proposed attack indeed works against vanilla models. Therefore, we cannot arrive at conclusions like "adv-trained DEQs have better robustness than standard models" without seeing the results on the vanilla-trained DEQs.

* Some terms are not clearly defined and confusing to readers. For example, no specific names are given to the two approaches on intermediate gradient estimation proposed in Sec. 4.1 and 4.2. I hypothesize that the terms "exact-trained" and "unrolling-trained" will sometimes refer to the two approaches respectively, but they are also used to refer to the two previous works for training DEQs.

---

> ### Author Response · Authors · 2022-08-02
> **Thank you for your valuable review**
>
> Thank you for your valuable review and suggestions. We have uploaded a revision of our paper.
>
> ***Question 1: Paper logic and the attacks against vanilla-trained DEQs***
>
> We appreciate the constructive suggestion and we train a vanilla DEQ on CIFAR-10 following the recipe in [1]. We use the ready-made PGD-10 to attack the vanilla-trained DEQ model. The clean accuracy (\%) and robust accuracy (\%) of each state $z_n$ are shown below:
>
> | State       | z1    | z2    | z3    | z4    | z5    | z6    | z7    | z8    |
> |-------------|-------|-------|-------|-------|-------|-------|-------|-------|
> | Clean Acc.  | 38.81 | 82.62 | 89.63 | 91.77 | 92.08 | 92.29 | 92.39 | 92.53 |
> | Robust Acc. | 2.00  | 0.00  | 0.00  | 0.00  | 0.00  | 0.00  | 0.00  | 0.00  |
>
> As observed, the ready-made PGD-10 already has a dramatic effect in attacking all the states against vanilla-trained DEQs.
>
> We proceed to apply the proposed attacks and defense strategies. We follow Section 5.2 to determine the optimal early exit as state z1. The results are shown below ("SA" stands for the attacks with "simultaneous adjoint" and "UI" stands for the attacks with "unrolled intermediates"; best viewed in the new Table 13):
>
> | Defense  | Clean |      SA      |      SA      |    SA    |    UI    |      UI       |    UI    |
> |----------|-------|--------------|--------------|----------|----------|---------------|----------|
> |          |       | Final        | Intermediate | Ensemble | Final    | Intermediate  | Ensemble |
> | Final    | 92.53 | 8.90         | 11.45        | 3.69     | 0.00     | 0.00          | 0.00     |
> | Early    | 38.81 | 6.08         | 4.54         | 3.42     | 2.00     | 2.94          | 1.31     |
> | Ensemble | 87.31 | 9.12         | 6.39         | 3.48     | 0.00     | 0.00          | 0.00     |
>
> We can see that all the proposed attacks can defeat a vanilla-trained DEQ. As the white-box robustness of DEQs is assessed by the strongest defense under all attacks (minimum over all columns in a row, then maximum over the minimum of the rows), the white-box robustness of the vanilla DEQ is 1.31\% with a 38.81\% clean accuracy using the early-state defense. When using the final-state and the ensemble-state defense, the robustness is 0.00\%. We include detailed discussions in Lines 267~269 and Appendix E of the revised paper.
>
> ***Question 2: The actual meaning of the terms "exact-trained" and "unrolling-trained", and their relationship with the two different intermediate gradient estimation techniques (without specific names) in Sections 4.1 & 4.2***
>
> The term "exact-trained" means using the exact gradient (Eqs. (2) and (3)) in the DEQ adversarial training procedure (both to generate adversaries with $\frac{\partial L}{\partial x}$ in PGD-AT and to optimize for parameters with $\frac{\partial L}{\partial \theta}$), while the term "unrolling-trained" means using the unrolling-based phantom gradient (Eqs. (4), (5), and (6)) during the DEQ adversarial training procedure. These two terms refer to the two previous works for training DEQs. We more clearly clarify their definitions in Lines 202~204 of the revised paper.
>
> We use the terms "simultaneous adjoint" and "unrolled intermediates" to refer to the two intermediate gradient estimation techniques in Section 4. These terms appeared in the headers of Tables 2 and 3; We further name these two methods explicitly in Section 4, Lines 142\~143 and Lines 168\~169 of the revised paper. As illustrated in Figure 1, these two techniques estimate the gradient w.r.t. some intermediate state ($z_n$ , $1<=n<=N$), while the exact/unrolling-based phantom gradient is defined w.r.t. the final state ($z^* = z_N$). The two intermediate gradient estimation techniques are integrated into the construction of the white-box adaptive attacks in Section 5.1.
>
> As for their relationship in experiments, "exact/unrolling-trained" refers to **training strategies**, while "simultaneous adjoint/unrolled intermediates" refers to **evaluation strategies**. More specifically, we first apply the exact or unrolling training strategy to obtain the adversarially-trained DEQs, then use adaptive attacks with both of the intermediate gradient estimation methods to perform evaluation (as shown in Tables 2 and 3).
>
> ***Question 3: The discussion on the limitation of the paper***
>
> Indeed, (adversarially) training DEQs on large-scale datasets is much more non-trivial compared to standard DL models. Nevertheless, recent progress [1,2] has shown that DEQ models can work well on large-scale vision tasks including ImageNet and Cityscapes. Given this, we could apply, e.g., FastAT [3] to DEQs, which preserves the scalability of our methods. In the updated version of the paper, we have added more discussions in Appendix I.
>
> ***References:***
>
> [1] Stabilizing Equilibrium Models by Jacobian Regularization. ICML 2021
>
> [2] Multiscale Deep Equilibrium Models. NeurIPS 2020
>
> [3] Fast is Better than Free: Revisiting Adversarial Training. ICLR 2020

---

> > ### Comment · Reviewer_LJfG · 2022-08-09
> > **Thank you for the experiments**
> >
> > Thank you for the extra experiments and explanations. The extra results solved my concern and I would raise my score.

---

> > > ### Author Response · Authors · 2022-08-09
> > > **Thank you again**
> > >
> > > Thank you for updating the score! Your feedback really helped us improve our work. We will further revise our paper with the added experiments and discussions.

---

### Official Review · Reviewer_L5vH · 2022-07-12

**Rating:** 5
**Confidence:** 2
**Soundness:** 3 good
**Presentation:** 3 good
**Contribution:** 3 good

**Summary:**

This paper evaluated the robustness of the general deep equilibrium model (DEQ) in the traditional white-box attack-defense setting. The authors first pointed out the challenges of training robust DEQs. Then they developed a method to estimate the intermediate gradients of DEQs and integrate them into the adversarial attack pipelines.

**Questions:**

1. In Table 3, I noticed that for many models AutoAttack's performance is worse than PGD10, which surprised me. I was wondering if this is something special for DEQs. More explanation on this is welcome.

**Limitations:**

The authors adequately addressed the limitations and potential negative social impact of their work.

**Strengths And Weaknesses:**

Strength:
1. General DEQs' robustness is not well studied in the literature and this paper provides the first study in this area.
2. The intermediate gradient estimation methods proposed are interesting, especially the first one inspired by the adjoint process in neural ODE models. I am wondering if such methods can be applied to attacks on the regular deep neural networks.

Weakness:
1. I found the intro to DEQs a little bit hard to follow. I spent a lot of time on the formulation of the problem. I think if the authors want to present this work in the adversarial robustness community, a friendlier intro in Section 2 may be useful.
2. The time complexity of "simultaneous adjoint in the forward pass" proposed in 4.1 seems high, especially with a Jacobian inverse (even with low-rank approximation). Some complexity analysis and running time report may be useful.

---

> ### Author Response · Authors · 2022-08-02
> **Thank you for the supportive review**
>
> Thank you for the supportive review and kind suggestions. We have uploaded a revision of our paper.
>
> ***Question 1: A friendlier introduction to DEQs in Section 2***
>
> We will certainly improve and polish the introduction to DEQs in the revised paper. Intuitively, DEQs propose to directly solve for the latent representation equilibrium $z^*$, which is defined by a fixed-point equation as $z^* = f_\theta(z^*; x)$. Here $f_\theta$ represents a single-layer transformation parametrized by $\theta$. For DEQs, the forward pass is the iterative root-finding process searching for $z^*$, which is implemented via black-box solvers like Broyden's method; similarly, the backward pass differentiating through the loss function $L$ yields another fixed-point structure, which can be treated as the root-finding process for $u^*$ described as $u^* = \left(\frac{\partial f_\theta(z^*, x)}{\partial z}\right) u^* + \frac{\partial L(z^{*}, y)}{\partial z}$ and solved by another black-box solver. Compared with deep networks, DEQs require only O(1) memory and have a broad range of applications from language modeling, image recognition, graph modeling to optical flow estimation.
>
>
> ***Question 2: Complexity analysis and running time report for simultaneous adjoint in the forward pass***
>
> The time complexity of approximating the Jacobian inverse in the simultaneous adjoint is equivalent to that in the original forward pass, since we directly **reuse** the approximated Jacobian inverse $B_n$ in the forward calculation of $z_n$ when calculating the adjoint state $u_n$. This is different from the original DEQ design, where the forward and the backward passes are decoupled by separate fixed-point solvers, and consequently different $B_n$'s need to be maintained separately.
>
> Concretely, compared with the original forward pass (Eqs. (7) and (8)), the simultaneous adjoint calculation augments it with Eqs. (9) and (10). The time complexity of Eq.(10) is equivalent to that of Eq.(7), while Eq. (9) calculates the residual of Eq.(3) as
>
> $$ u^* = \left(\frac{\partial f_\theta(z^*, x)}{\partial z}\right) u^* + \frac{\partial L(z^{*}, y)}{\partial z}, $$
>
> at $u^* = u_n$ and $z^* = z_n$. As Eq. (3) defines the fixed-point equation of the backward pass in an original DEQ, the time complexity of Eq. (9) is equivalent to just the **residual evaluation** when solving for the exact gradient in an original DEQ.
>
> In practice, the running time comparison using the adaptive PGD-10 attack with **(a)** the final adjoint state in "simultaneous adjoint" and **(b)** the unrolled final state in "unrolled intermediates" is
>
> | Type of attack          | Time used per batch (ms) |
> |-------------------------|--------------------------|
> | Simultaneous Adjoint    | 6,479                    |
> | Unrolled Intermediates  | 5,369                    |
>
> We can see that the computational burden introduced from Eq.(9) and Eq. (10) does not take the majority of the running time. We have included these discussions in the added Appendix G.2 and referred to them in Line 259 in the updated main paper.
>
> ***Question 3: Why AutoAttack's performance is worse than PGD10 for DEQs***
>
> AutoAttack is stronger than PGD-10 under the condition that the model gradients are accurately accessible (without gradient obfuscation), which is usually true for deep networks by automatic differentiation. However, as explained in Lines 277~289, the model gradients of DEQs at the intermediate states can only be estimated (because of the black-box fixed-point solvers used in DEQs) and are not completely accurate. Therefore, note that the APGD-CE / APGD-T methods in AutoAttack use adaptive step size, which facilitates the constructed perturbation to overfit the (inaccurate) gradients. In contrast, PGD-10 applies constant step size, which is usually less powerful against deep networks, but happens to act as a regularization in the case of DEQs, especially when the defense strategies leverage an output state (final, early, or ensemble) different from the state that the intermediate gradient is estimated at.

---

### Author Response · Authors · 2022-08-05
**Looking forward to further feedbacks**

Dear Reviewers,

Thank you again for your valuable comments and suggestions, which are really helpful for us. We have posted responses to the detailed concerns.

We totally understand that this is a quite busy period, since the reviewers may be responding to the rebuttal of other assigned papers.

We deeply appreciate it if you can take some time to return further feedback on whether our responses solve your concerns. If there are any other comments, we will try our best to address them.

Best,

The authors

---

### Meta-Review · Area_Chair_23QV · 2022-08-28

**Recommendation:** Accept
**Confidence:** Certain

**Metareview:**

This paper studies the empirical robustness of the general deep equilibrium model (DEQ) in the traditional white-box attack-defense setting.  As the topic is under-explored in the literature, the authors first pointed out the challenges of training robust DEQs. Then, they developed a method to estimate the intermediate gradients of DEQs and integrate them into the adversarial attack pipelines. The authors did a good job to address the reviewers' concerns in the author-reviewer discussion phase, and at the end, all reviewers unanimously support the acceptance. Although AC sees some limitations, e.g., limited advantages of using robust DEQs over deep CNNs, scalability to large-scale datasets and training instability, AC thinks the merits of this paper outweigh them: this paper can be a useful guideline when researchers pursue the under-explored problem in the future. Hence, AC recommends acceptance.

**Award:**

No

---

### Decision · Program_Chairs · 2022-09-14

Accept